# Task-Oriented Diffusion Inversion for High-Fidelity Text-based Editing

A WOMAN IN A JACKET STANDING IN THE RAIN → A WOMAN IN A BLOUSE STANDING IN THE RAIN

A GOLD PLATED BOWL FILLED WITH FRUIT → A GOLD PLATED BOWL FILLED WITH CANDY

A HOUSE WITH LIGHTNING AND RAIN ON IT → A HOUSE WITH RAIN ON IT

A WOMAN IN SUNGLASSES ... → WATERCOLOR STYLE OF A WOMAN IN SUNGLASSES ...

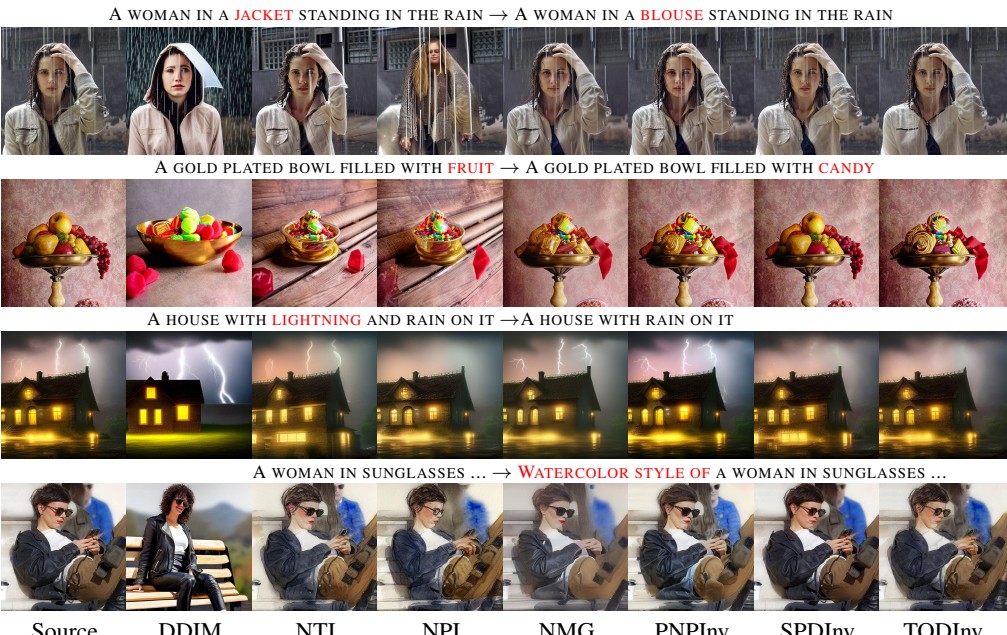

| Source | DDIM | NTI | NPI | NMG | PNPInv | SPDInv | TODInv |

Figure 1: Our TODInv framework seamlessly integrates the inversion process with editing tasks, enabling diverse high-fidelity text-guided edits such as object replacement, object removal, and stylization. The edited images not only retain the original background but also perfectly align with the target prompts.

## Abstract

Recent advancements in text-guided diffusion models have unlocked powerful image manipulation capabilities, yet balancing reconstruction fidelity and editability for real images remains a significant challenge. In this work, we introduce **T**ask-**O**riented **D**iffusion **I**nversion (**TODInv**), a novel framework that inverts and edits real images tailored to specific editing tasks by optimizing prompt embeddings within the extended $\mathcal{P}^*$ space. By leveraging distinct embeddings across different U-Net layers and time steps, TODInv seamlessly integrates inversion and editing through reciprocal optimization, ensuring both high fidelity and precise editability. This hierarchical editing mechanism categorizes tasks into structure, appearance, and global edits, optimizing only those embeddings unaffected by the current editing task. Extensive experiments on benchmark dataset reveal TODInv's superior performance over existing methods, delivering both quantitative and qualitative enhancements while showcasing its versatility with few-step diffusion model.

## 1 Introduction

Text-guided diffusion models (Rombach et al., 2022; Xue et al., 2024; Saharia et al., 2022) have achieved significant success in synthesizing realistic images due to their controllability and diversity. Leveraging these effective text-guided diffusion models, numerous works have explored the

generative priors of pre-trained diffusion models and successfully applied these capabilities to various downstream tasks (Zhao et al., 2023; Qi et al., 2023; Wu et al., 2023; Chen et al., 2023; Ji et al., 2023; Baranchuk et al., 2022), particularly in text-driven image and video editing (Wu et al., 2023; Chai et al., 2023; Qi et al., 2023; Tumanyan et al., 2023; Hertz et al., 2023; Khachatryan et al., 2023; Saharia et al., 2022; Cao et al., 2023). These technologies enable users to edit images according to their desires via text modification.

When editing a real image $x_0$, many text driven image editing methods (Hertz et al., 2023; Cao et al., 2023; Tumanyan et al., 2023; Parmar et al., 2023) require to invert $x_0$ into the latent space of a pre-trained diffusion model to obtain the corresponding latent codes $\{z_t\}_{t=T}^1$, which is the inverse process of the diffusion model's sampling procedure. There are two key aspects to this task: the fidelity of the reconstruction and the editability of the latent codes (Garibi et al., 2024; Pan et al., 2023). A naive approach to this task is Denoising Diffusion Implicit Models (DDIM) inversion (Dhariwal & Nichol, 2021; Song et al., 2021), which reverses the source image according to the DDIM sampling schedule. However, applying DDIM inversion to text-guided diffusion models often fails due to Classifier Free Guidance (CFG) (Ho & Salimans, 2022), which uses conditional text as input and magnifies the approximation error.

To eliminate the approximation error in DDIM inversion, many works (Sohl-Dickstein et al., 2015; Mokady et al., 2023; Han et al., 2024; Miyake et al., 2023) align the differences between conditional and unconditional trajectories to ensure that the source image is faithfully reconstructed. In addition to aligning the two trajectories directly, several works reduce the approximation error at each timestep by optimizing the latent codes. Specifically, AIDI (Pan et al., 2023), FPI (Meiri et al., 2023), and ReNoise (Garibi et al., 2024) introduce a fixed-point iteration process in each inversion step to obtain accurate latent codes. Furthermore, SPDInv (Li et al., 2024) optimizes latent codes directly based on the difference between two adjacent latent codes. Despite the progress made in fidelity reconstruction, the optimized latent codes often exhibit reduced editability (Garibi et al., 2024; Parmar et al., 2023).

To achieve an ideal balance between reconstruction fidelity and editability, we argue that these two tasks must be intrinsically linked and not treated separately. The inversion process should be highly tailored to the specific editing task at hand. This necessity arises because different edited outputs are modified at varying sampling steps or layers of a diffusion model (Patashnik et al., 2023; Liew et al., 2022). As a result, for a given real image, it is crucial to obtain distinct optimal latent codes corresponding to each editing output.

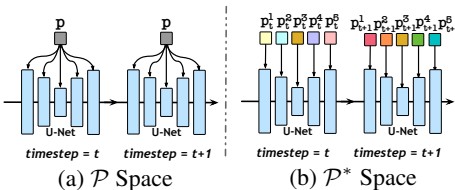

(a) $\mathcal{P}$ Space   (b) $\mathcal{P}^*$ Space

Figure 2: Illustration of original and extended prompt spaces.

Furthermore, we discern that various text-driven image editing tasks can be broadly categorized into three distinct classes: structure editing, appearance editing, and structure-appearance (i.e., global) editing. The modulation of appearance and structure is controlled by different layers within the U-Net architecture during the diffusion process. This leads us to assert that varying levels of editing should correspondingly activate different tiers of text embeddings. These insights motivate the creation of an inversion framework that dynamically integrates edit instructions in a hierarchical manner, thereby ensuring both high fidelity and precise editability. In this paper, we propose a novel **T**ask-**O**riented **D**iffusion **I**nversion (**TODInv**) framework designed to invert and edit real images tailored to specific editing tasks. Our approach focuses on inverting to prompt embeddings in individual layers. This method represents the input real image through a sequence of prompt embeddings, which can be effectively edited in downstream applications. In particular, we optimize the prompt embeddings within the extended prompt embedding space $\mathcal{P}^*$ (Alaluf et al., 2023). As illustrated in Fig. 2, unlike the original prompt space $\mathcal{P}$, which shares the same embedding across different time steps and U-Net layers, the $\mathcal{P}^*$ space employs distinct embeddings at different layers and time steps. This extended space integrates the disentanglement and expressiveness of time and space, benefiting our inversion in two key aspects:

i) The expressiveness of this latent space facilitates the minimization of inversion errors, significantly enhancing reconstruction accuracy.

ii) Compared to the original $\mathcal{P}$ space, $\mathcal{P}^*$ space is more disentangled, which allows for more precise optimization tailored to the specific editing type.

To obtain a faithful reconstruction tailored to the target editing task, we optimize only those prompt embeddings that are agnostic to the current editing, thereby minimizing approximation errors without compromising editability. We conduct extensive experiments on benchmark datasets utilizing various text-driven image editing technologies (Hertz et al., 2023; Cao et al., 2023; Tumanyan et al., 2023). As shown in Fig. 1, the experimental results indicate that our method outperforms existing diffusion inversion techniques in both quantitative and qualitative evaluations. Additionally, our method demonstrates strong performance with few-step diffusion models, further showcasing its versatility and effectiveness.

In summary, our contributions are as follows:

• We present TODInv, a novel diffusion inversion framework that seamlessly links and jointly optimizes inversion and editing processes, achieving both faithful reconstruction and high editability.
• We introduce a task-oriented prompt optimization strategy, categorizing various editing tasks into three types. For each class of editing, we minimize the approximation error by optimizing specific prompt embeddings that are irrelevant to the current editing.
• Extensive experiments on benchmark dataset demonstrate the effectiveness of our method over state-of-the-art techniques. Our inversion model also supports few-step diffusion models.

## 2 RELATED WORKS

**Image Editing via Diffusion Models.** Diffusion models (Rombach et al., 2022; Saharia et al., 2022; Ramesh et al., 2022) have made significant advancements in generating diverse and high-fidelity images guided by text prompts. Leveraging these powerful models, numerous works have harnessed their generative capabilities for text-driven image editing. For instance, Prompt-to-Prompt (P2P) (Hertz et al., 2023) manipulates attention modules in Stable Diffusion (Rombach et al., 2022) for localized and global edits. Plug-and-Play (PNP) (Tumanyan et al., 2023) adjusts spatial features and self-attention modules for fine-grained edits, while Pix2pix-Zero (Parmar et al., 2023) retains cross-attention maps for image-to-image translation. Recently, MasaCtrl (Cao et al., 2023) has enabled complex non-rigid editing by converting the self-attention module into mutual self-attention. Additionally, several works (Wu et al., 2023; Liu et al., 2024; Geyer et al., 2024; Zhang et al., 2024) have extended these methods to video editing. To apply these techniques to real images, inverting the images to the latent space of the diffusion model is a crucial first step.

**Inversion in Diffusion Models.** Early inversion methods for real image editing focused on Generative Adversarial Networks (GANs) (Xu et al., 2023; 2021; Creswell & Bharath, 2018; Abdal et al., 2019; 2020; Xia et al., 2023). The advent of diffusion models has shifted attention to diffusion-based inversion methods, which can be categorized into Denoising Diffusion Probabilistic Models (DDPM)-based (Huberman-Spiegelglas et al., 2024; Wu & De la Torre, 2023) and Denoising Diffusion Implicit Models (DDIM)-based approaches (Garibi et al., 2024; Dhariwal & Nichol, 2021; Song et al., 2021; Pan et al., 2023; Li et al., 2024; Meiri et al., 2023). DDPM-based methods leverage the denoising process but require a large number of inversion steps (Wu & De la Torre, 2023; Huberman-Spiegelglas et al., 2024). DDIM-based methods introduce a deterministic DDIM sampler for inversion. However, when CFG is used, DDIM inversion often fails to achieve high-fidelity reconstruction (Mokady et al., 2023). To address these issues, several works (Mokady et al., 2023; Han et al., 2024; Miyake et al., 2023) align the conditional and unconditional trajectories by optimizing the null text token or the prompt embedding. Concurrently, methods like EDICT (Wallace et al., 2023) and BDIA (Zhang et al., 2023a) introduce invertible networks for inversion. PNPInv (Ju et al., 2024) merges differences between reconstruction and editing branches, while NMG (Cho et al., 2024) utilizes spatial context from DDIM inversion for faithful editing. Despite these advancements, existing methods still suffer from approximation errors in DDIM inversion, as the process approximates latent $x_t$ using $x_{t-1}$. To eliminate these errors, techniques like AIDI (Pan et al., 2023), FPI (Meiri et al., 2023), and ReNoise (Garibi et al., 2024) introduce fixed-point iteration processes to optimize latent codes. SPDInv (Li et al., 2024) reformulates this iteration as a loss function. However, directly optimizing latent codes often results in reduced editability (Garibi et al., 2024; Parmar et al., 2023).

In contrast to existing solutions, our task-oriented inversion approach optimizes specific prompt embeddings in an extended prompt space for both inversion and editing, thereby avoiding the trade-off between faithful reconstruction and editability. While our method shares similarities with related

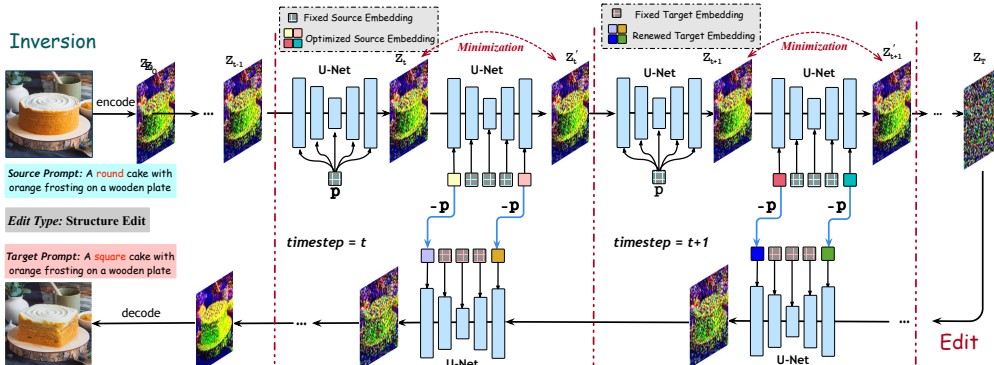

Figure 3: Overview of our TODInv. Given a real image, we first encode the image to the initial latent code $z_0$ using the encoder of Stable Diffusion. In timestep $t$, we get the latent code $z_t$ based on latent code $z_{t-1}$ and fixed source prompt embedding $p$ using Eq. 5, but bring the approximation error. Then we use $z_t$ to predict latent code $z_t'$ and minimize their distance by optimizing specific prompt embeddings according to the edit class. The final latent code $z_T$ can be cooperated with various editing methods, with the renewed the target prompts using Eq. 10 (the blue arrows)). *Note that only the structure of "*CAKE*" is edited in this example, which belongs to **structure edit**, We only optimize the appearance-related prompt embeddings (denoted by the colorful boxes without grids). For more detailed illustration on how to select the optimization layers, please see in Fig. 4.*

works (Mokady et al., 2023; Dong et al., 2023; Han et al., 2024) in prompt optimization, it distinguishes itself in two key aspects: 1) We optimize prompt embeddings to minimize approximation errors in the text-conditioned trajectory of DDIM inversion, rather than merely aligning null-text and text-conditioned trajectories. 2) Our approach specifically connects the inversion process to the editing tasks by optimizing prompt embeddings in the extended $\mathcal{P}^*$ space, focusing on embeddings irrelevant to the current editing task. This ensures high-fidelity reconstruction tailored to specific edits without compromising the ability to perform diverse and precise modifications.

**Extended Spaces of Diffusion Models.**  To better leverage the generative capabilities of diffusion models, several works have analyzed the latent space of these models. Voynov *et al.* (Voynov et al., 2023) extended the original prompt space to $\mathcal{P}+$ by using different embeddings for different U-Net layers, disentangling structure and appearance. Prospect (Zhang et al., 2023b) categorized denoising timesteps into style, content, and layout embeddings. NeTI (Alaluf et al., 2023) introduced a space-time space $\mathcal{P}*$ for personalized generation. Our work integrates temporal and layer-wise prompt spaces into a unified space, leveraging its expressiveness and disentanglement to achieve high-fidelity reconstruction and editability in diffusion inversion.

# 3 METHODOLOGY

## 3.1 PRELIMINARIES

In this section, we present the background of diffusion models and then analyze the approximation error in DDIM Inversion.

### 3.1.1 DIFFUSION MODELS

Diffusion models aim at mapping the random noise $z_T$ to a series latent code $\{z_t\}_{t=T}^1$, where $T$ is the number of timestep, and finally generate a clean image or latent code $z_0$, A diffusion model consists of a training process and a reverse inference process. To train a diffusion model, we add the noise $\epsilon \in \mathcal{N}(0, 1)$ to the real image $z_0$ to get the latent variable $z_t$ using follow equation:

$$z_t = \sqrt{\alpha_t} z_0 + \sqrt{1 - \alpha_t} \epsilon, \tag{1}$$

where $\alpha$ is the hyper-parameter. In a text-guided diffusion model, the text prompt embedding $p$ is conditioned on the network $\epsilon_\theta$ to predict the noise, and it is trained using the following equation:

$$\mathcal{L}_{\text{DM}} = \|\epsilon - \epsilon_\theta(z_t, p, t)\|_2^2. \tag{2}$$

During the inference, the clean image $z_0$ can be generated from random noise $z_T$ using deterministic DDIM sampler (Song et al., 2021) step by step:

$$z_{t-1} = \phi_t z_t + \psi_t \epsilon_\theta(z_t, p, t), \tag{3}$$

where $\phi_t$ and $\psi_t$ are sampler parameters, and $\phi_t = \frac{\sqrt{\alpha_{t-1}}}{\sqrt{\alpha_t}}, \psi_t = \sqrt{\alpha_{t-1}}\left(\sqrt{\frac{1}{\alpha_{t-1}} - 1} - \sqrt{\frac{1}{\alpha_t} - 1}\right)$.

### 3.1.2 DDIM INVERSION

Diffusion inversion is a reverse process of sampling, which aims to invert a clean image $z_0$ to the noise latent code $z_T$. According to Eq. 3, $z_T$ can be inverted from $z_0$ by following equation iteratively:

$$z_t = \frac{z_{t-1} - \psi_t \epsilon_\theta(z_t, p, t)}{\phi_t}. \tag{4}$$

However, directly computing $z_t$ using Eq. 4 is infeasible since the network $\epsilon_\theta(\cdot, \cdot)$ needs the $z_t$ as input. DDIM inversion assumes that the Ordinary Differential Equation (ODE) process can be reversed in the limit of infinitesimally small steps, and replace $z_t$ with $z_{t-1}$ for the noise prediction:

$$z_t \approx \frac{z_{t-1} - \psi_t \epsilon_\theta(z_{t-1}, p, t)}{\phi_t}. \tag{5}$$

This approximation error is introduced into every timestep of DDIM inversion, the accumulated errors decrease the reconstruction quality and editing ability (Pan et al., 2023; Meiri et al., 2023; Li et al., 2024; Garibi et al., 2024). Moreover, in the recent few-step diffusion models (Luo et al., 2023a;b; Sauer et al., 2023; Song et al., 2023), the approximation error between $z_{t-1}$ and $z_t$ is significantly large, DDIM inversion suffers worse performance on reconstruction (Garibi et al., 2024).

### 3.2 APPROXIMATION ERROR MINIMIZATION

For minimizing the approximation error in the DDIM inversion, existing works (Pan et al., 2023; Meiri et al., 2023; Garibi et al., 2024; Li et al., 2024) optimize the latent code $z_t$ directly in each timestep. In those works, the fidelity reconstruction can be guaranteed, but compromises the editability.

Instead, we optimize the prompt embeddings, rather than original latent codes. A naive solution is optimizing the prompt embedding in the original prompt space $\mathcal{P}$. In timestep $t$, we first get the latent code $z_t$ based on $z_{t-1}$ with DDIM inversion (using Eq. 5), then we take the obtained $z_t$ and prompt embedding $p$ to predict another latent code $z_t'$, and we minimizing the distance between the input and output codes by optimizing prompt embedding $p$. The above description can be represented as:

$$z_t' = \frac{z_{t-1} - \psi_t \epsilon_\theta(z_t, p, t)}{\phi_t}, \tag{6}$$

$$p^* = \arg\min_p \|z_t' - z_t\|_2^2. \tag{7}$$

However, optimizing prompt embedding directly has two drawbacks. Firstly, for the original space $\mathcal{P}$, a single text embedding is injected to networks regardless of timesteps and layers of U-Net, the optimization of this shared text embedding limits the minimization of Eq. 7 across different timesteps. Secondly, as indicated by the customized diffusion works (Ruiz et al., 2023; Xu et al., 2024), the optimized $p^*$ also encodes the image context after optimization, leading to the decreased editability.

### 3.3 TASK-ORIENTED PROMPT OPTIMIZATION

For achieving the high fidelity reconstruction meanwhile preserving the editability, we argue that the inversion process should be oriented to the edit task, as a universally optimal latent code adept at both faithful reconstruction and diverse editing tasks is unattainable. We observe various image editing tasks can be broadly categorized into three classes: structure editing ("EDIT A ROUND YELLOW CAKE TO SQUARE YELLOW CAKE"), appearance editing ("EDIT A ROUND YELLOW CAKE TO ROUND RED CAKE"), and global editing ("EDIT A ROUND YELLOW CAKE TO SQUARE RED CAKE"). On the other hand, It's evidenced that the structure and appearance are modulated by different layers' prompts (Alaluf et al., 2023; Voynov et al., 2023). This leads us to assert that varying levels of editing should correspondingly different layers of text embeddings.

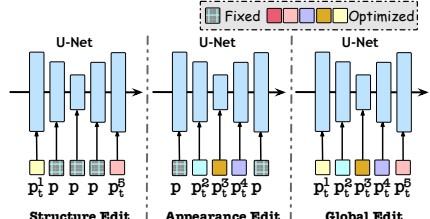

Figure 4: We categorize all kinds of editing tasks into three classes and divide different layers of U-Net into structure and appearance layers according to their resolutions. For each kind of editing, we only optimize the prompt embeddings that are irrelevant to this editing.

In our task-oriented inversion, to avoid embedding the content of specific prompts which decreases the editability after minimizing the approximation error, we only optimize the prompt embeddings that are irrelevant to current editing (see in Fig 4). For example, for the appearance editing, we only update those embeddings related to the structures. As the appearance-related prompt embeddings are kept fixed, the editability will not be decreased. We chose the extended prompt space $\mathcal{P}^*$ proposed by (Alaluf et al., 2023) for optimization, as it is evidenced to be more expressive and disentangled.

Let $p_t^i \in \mathcal{P}^*$ denotes the prompt embedding injected to the $i$ resolution layer of U-Net at $t$ timestep, we follow (Alaluf et al., 2023; Voynov et al., 2023) that class different layer prompt embeddings into two groups according to the resolution: the structure prompt set in the low-resolution layers: $P_t^{str} = [p_t^i, i \in low\ res\ layers]$, and the appearance prompt set controls the high-resolution layers: $P_t^{app} = [p_t^j, j \in high\ res\ layers]$, we first get the latent code $z_t'$ by replacing $p$ with $[P_t^{str}, P_t^{app}]$ in Eq. 6:

$$z_t' = \frac{z_{t-1} - \psi_t \epsilon_\theta(z_t, [P_t^{str}, P_t^{app}], t)}{\phi_t}. \tag{8}$$

Then, for the appearance-related editing, we optimize the irrelevant structure embeddings set $P_t^{str}$, and vice versa. For the global editing, we optimize all the prompt embeddings, which can be represented as:

$$P_t^* = \begin{cases} \arg\min_{P_t^{app}} \|z_t - z_t'\|_2^2 & if\ \text{structure editing;} \\ \arg\min_{P_t^{str}} \|z_t - z_t'\|_2^2 & elif\ \text{appearance editing;} \\ \arg\min_{P_t^{str}, P_t^{app}} \|z_t - z_t'\|_2^2 & else\ \text{global editing.} \end{cases} \tag{9}$$

We follow (Li et al., 2024; Dong et al., 2023) that set the maximum optimization steps as $K$ in each timestep, meanwhile, we also set a threshold $\delta$ to control the termination of the optimization process. By feeding the latent code $z_t$ with the optimized prompt embeddings $P_t^*$ to the U-Net, with the DDIM sampler, the original image can be reconstructed faithfully. More importantly, with task-oriented optimization, the editability will not be decreased. If the same image undergoes multiple types of edits during iterative editing, we choose global editing for optimization. This is because applying different edit categories requires optimizing prompt embeddings across all layers, similar to the global editing category.

During the editing, we leverage the difference between the original and optimized embeddings on the target prompt $P_t^{target}$, that is:

$$\tilde{P}_t^{target} = P_t^* - P_t + P_t^{target}, \tag{10}$$

where $\tilde{P}_t^{target}$ is the renewed target prompt embedding. Incorporated with various text-driven image editing methods (Cao et al., 2023; Hertz et al., 2023; Tumanyan et al., 2023), we can edit the real image with target prompt.

## 4 EXPERIMENTS

### 4.1 EXPERIMENTAL SETTINGS

**Dataset.** To evaluate the effectiveness of our hierarchical inversion, we conduct experiments on the PIE-Bench dataset proposed by PNPInv (Ju et al., 2024), which consists of 700 images with 9 editing types. Each image is annotated with the source and target prompts. Meanwhile, this dataset also provides the editing region masks for evaluation.

**Evaluation Metrics.** We follow PNPInv (Ju et al., 2024) which uses several metrics to evaluate our method. We first use the **Structure Distance** assessed by DINO score (Caron et al., 2021) to evaluate the structure distance between original and edited images. Note that this metric cannot be used to evaluate structural edits, as neither higher nor lower values effectively reflect the desired changes. However, we follow the official evaluation proposed by (Ju et al., 2024), which adopts a "lower is better" approach for the entire dataset. We also introduce several metrics to evaluating the background preservation, which includes **PSNR**, **LPIPS** (Zhang et al., 2018), **MSE**, and **SSIM** (Wang et al., 2004). Those metrics are calculated on the unedited regions, which are defined by the PIE-Bench dataset. Additionally, we introduce CLIP Similarity (Wu et al., 2021) to evaluate the text-image consistency between edited images and corresponding target editing text prompts both on the whole image and edited regions. At last, we introduce the **Inference Times** to evaluate different methods' inversion time cost on one image.

**Image Editing Methods.** We cooperate with various inversion methods with four text-guided image editing methods, including P2P (Hertz et al., 2023), MasaCtrl (Cao et al., 2023), PNP (Tumanyan et al., 2023), and Pixel-Zero (Parmar et al., 2023). Note that not all inversion method provides the source code with MasaCtrl, PNP, and Pixel-Zero editing, we only compare all methods with P2P editing. For the few-step diffusion models, we follow ReNoise that edits the images by replacing the target word directly.

### 4.2 IMPLEMENTATION DETAILS

We implement the proposed method in PyTorch on a PC with Nvidia GeForce RTX 3090. We use Stable Diffusion V1.4 as our main text-guided diffusion model and set the CFG scale as 7.5. We use the AdamW optimizer (Loshchilov & Hutter, 2019) with the learning rate is set to be 0.001. We categorize 9 editing types in PIE-Bench dataset into three classes. Particularly, the structure editing contains Add Object, Delete Object, Change Content, and Change Pose. The appearance editing contains Change Color, Change Material, and Change Style, and the global editing only contains Change Background. Additionally, the U-Net of diffusion model has 4 resolution layer scales, i.e., $64 \times 64$, $32 \times 32$, $16 \times 16$, and $8 \times 8$. Inspired by (Voynov et al., 2023), we take the resolutions of $64 \times 64$ and $32 \times 32$ as appearance layers, and $16 \times 16$, $8 \times 8$ as structure layers. We set the maximization optimization steps $K=10$, and follow (Mokady et al., 2023; Li et al., 2024) set threshold $\delta$ as $5e^{-6}$.

### 4.3 QUANTITATIVE COMPARISON

We present the quantitative comparisons with state-of-the-art methods based on various text-guided image editing methods in Tab. 1, we can see that our TODInv outperforms competitors with various editing techniques on most of the evaluation metrics. SPDInv is beyond our method on some reconstruction metrics, but it has a worse editability. As discussed in Sec.3.2, that is because it optimizes the latent code directly for the faithful reconstruction, but ignores the important editing task, the same conclusion also can be drawn from Fig. 1 and Fig. 5, as it always failed on image editing. Thanks to our task-oriented prompt optimization, our method achieves faithful reconstruction and high editability performance. On the other hand, our method is more efficient than optimization works, because we optimize prompt embedding in the expressive $\mathcal{P}^*$ space, which is easier for optimization.

### 4.4 QUALITATIVE COMPARISON

The qualitative comparison with various inversion methods based on P2P (Hertz et al., 2023) edit can be seen in Fig. 1 and Fig. 5. We can see that the edited images obtained by DDIM always present

Table 1: Qualitative comparisons with related works using various text-guided editing methods.

| Method | | Structure | Background Preservation | | | | CLIP Similarity | | Times(s) ↓ |
|---|---|---|---|---|---|---|---|---|---|
| Inverse | Editing | Distance$_{\times 10^3}$ ↓ | PSNR ↑ | LPIPS$_{\times 10^3}$ ↓ | MSE$_{\times 10^4}$ ↓ | SSIM$_{\times 10^2}$ ↑ | Whole ↑ | Edited ↑ | |
| DDIM | P2P | 69.43 | 17.87 | 208.80 | 219.88 | 71.14 | 25.01 | **22.44** | **11.55** |
| NTI | P2P | 13.44 | 27.03 | 60.67 | 35.86 | 84.11 | 24.75 | 21.86 | 137.54 |
| NPI | P2P | 16.17 | 26.21 | 69.01 | 39.73 | 83.40 | 24.61 | 21.87 | 11.75 |
| StyleD | P2P | 11.65 | 26.05 | 66.10 | 38.63 | 83.42 | 24.78 | 21.72 | 382.98 |
| AIDI | P2P | 12.16 | 27.01 | 56.39 | 36.90 | 84.27 | 24.92 | 20.86 | 87.21 |
| FPI | P2P | 14.71 | 26.61 | 61.97 | 37.64 | 83.52 | 23.93 | 21.35 | 11.75 |
| NMG | P2P | 26.64 | 25.38 | 88.31 | 112.77 | 81.73 | 24.90 | 22.16 | 16.71 |
| ProxEdit | P2P | 8.80 | 28.31 | 44.13 | 25.72 | 85.74 | 24.15 | 21.36 | 11.75 |
| PNPInv | P2P | 11.65 | 27.22 | 54.55 | 32.86 | 84.76 | 25.02 | 22.10 | 19.94 |
| SPDInv | P2P | 8.81 | **28.60** | 36.01 | **24.54** | 86.23 | 25.26 | - | 27.04 |
| TODInv | P2P | **8.37** | 28.39 | 39.86 | 25.71 | **86.04** | **25.47** | 21.91 | 21.02 |
| DDIM | MasaCtrl | 28.38 | 22.17 | 106.62 | 86.97 | 79.67 | 23.96 | 21.16 | **11.55** |
| AIDI | MasaCtrl | 55.93 | 19.25 | 177.57 | 178.13 | 75.58 | 24.01 | 21.07 | 87.21 |
| NMG | MasaCtrl | 40.54 | 20.35 | 127.85 | 135.17 | 77.52 | 24.56 | 21.33 | 16.71 |
| ProxEdit | MasaCtrl | 21.28 | 23.81 | 85.52 | 66.47 | 81.62 | 23.60 | 20.94 | 11.75 |
| PNPInv | MasaCtrl | 24.70 | 22.64 | 87.94 | 81.09 | 81.33 | 24.38 | **21.35** | 19.94 |
| SPDInv | MasaCtrl | **20.48** | 24.12 | 71.74 | 64.77 | 82.54 | 24.61 | - | 27.04 |
| TODInv | MasaCtrl | **19.39** | **24.36** | **70.17** | **62.27** | **82.95** | **24.74** | 21.20 | 21.02 |
| DDIM | PNP | 28.22 | 22.28 | 113.33 | 83.51 | 79.00 | 25.41 | 22.55 | **11.55** |
| AIDI | PNP | 25.36 | 23.11 | 98.10 | 78.19 | 80.57 | 25.03 | **22.70** | 87.21 |
| PNPInv | PNP | 24.29 | 22.46 | 106.06 | 80.45 | 79.68 | 25.41 | 22.62 | 19.94 |
| SPDInv | PNP | **15.58** | **26.72** | 91.55 | **34.69** | 82.04 | 25.14 | - | 27.04 |
| TODInv | PNP | 21.06 | 25.13 | **78.49** | 50.16 | **82.83** | **26.08** | 22.50 | 21.02 |
| DDIM | P2P-Zero | 61.68 | 20.44 | 172.22 | 144.12 | 74.67 | 22.80 | 20.54 | **11.55** |
| PNPInv | P2P-Zero | **49.22** | **21.53** | **138.98** | **127.32** | **77.05** | 23.31 | 21.05 | 19.94 |
| TODInv | P2P-Zero | 49.86 | 21.34 | 139.47 | 134.66 | 76.91 | **24.19** | **21.15** | 21.02 |
| DDIM[†] | ReNoise | 216.17 | 14.52 | 319.53 | 464.16 | 54.30 | 21.17 | 18.38 | **0.56** |
| ReNoise[†] | ReNoise | 107.56 | 15.60 | 271.39 | 704.96 | 62.48 | 25.64 | 23.64 | 2.56 |
| TODInv[†] | ReNoise | **86.91** | **17.81** | **194.00** | **224.86** | **65.15** | **26.36** | **23.83** | 4.02 |

[†] use SDXL-Turbo as base model

an inconsistent background or structure with the source images, as pointed out by NTI (Dong et al., 2023), that is aroused by the CFG used in the sampling process.

Besides, all methods fail to replace the "JACKET" with "BLOUSE" in $1_{st}$ sample of Fig. 1 except ours, which indicates the effectiveness of our model in object replacement. The same conclusion also can be drawn from the $1_{st}$ sample of Fig. 5, as none of competitors can remove the "SNOW" on the fox's face. By disentangling the structure and appearance editing in the $\mathcal{P}^*$ space, our method is also skilled at changing the style of images, such as stylizing real images into "WATERCOLOR". We notice that SPDInv, AIDI, and FPI fail to replace the "BREAD" with "MEAT" in the $2_{nd}$ sample of Fig. 5, that is because all of them optimize the latent code for the faithful reconstruction, but reduces the editability. By minimizing the approximation error in each inversion timestep with a specific layer's prompt optimization, our method not only preserves the source background and structure but also supports various edits. For more qualitative comparisons using other editing methods, please see the supplementary material.

## 4.5 ABLATION STUDIES

In this section, we conduct an ablation experiment to analyze different choices in our TODInv. We first analyze the effectiveness of optimization in extended prompt space $\mathcal{P}^*$. Particularly, we develop three variants: 1) *Opti.* in $\mathcal{P}$, we optimize the prompt embedding in the original prompt

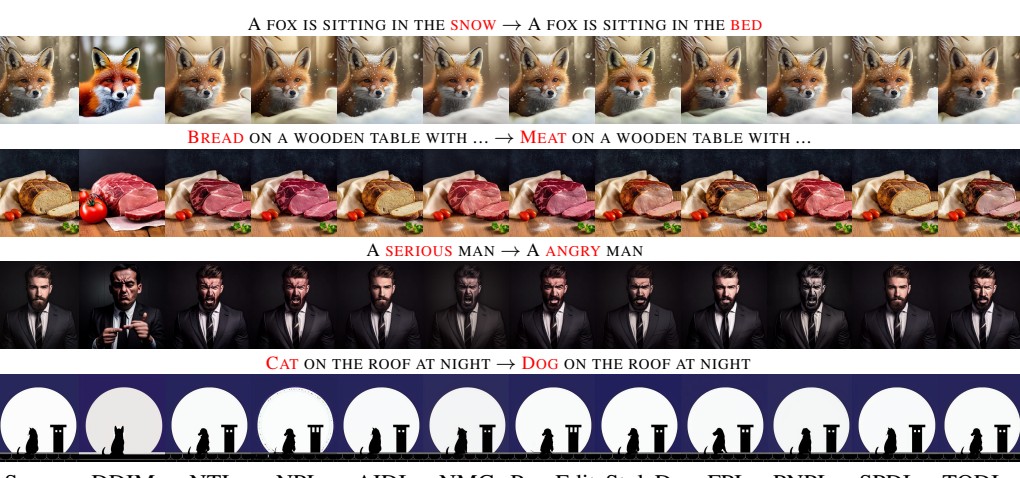

A FOX IS SITTING IN THE SNOW → A FOX IS SITTING IN THE BED

BREAD ON A WOODEN TABLE WITH ... → MEAT ON A WOODEN TABLE WITH ...

A SERIOUS MAN → A ANGRY MAN

CAT ON THE ROOF AT NIGHT → DOG ON THE ROOF AT NIGHT

Source  DDIM  NTI  NPI  AIDI  NMG  ProxEdit  StyleD  FPI  PNPInv  SPDInv  TODInv

Figure 5: Qualitative comparison with various inversion methods using P2P editing method.

Table 2: Qualitative comparisons with various variants using P2P editing.

| Variant | Structure | Background Preservation | | | | CLIP Similarity | | Times(s) ↓ |
|---|---|---|---|---|---|---|---|---|
| | Distance$_{\times 10^3}$ ↓ | PSNR ↑ | LPIPS$_{\times 10^3}$ ↓ | MSE$_{\times 10^4}$ ↓ | SSIM$_{\times 10^2}$ ↑ | Whole ↑ | Edited ↑ | |
| *Opti.* in $\mathcal{P}$ | 36.16 | 21.62 | 121.03 | 103.34 | 78.10 | 25.51 | 22.28 | 21.02 |
| *Opti.* in $\mathcal{P}_t$ | 35.93 | 21.71 | 120.47 | 102.25 | 78.15 | 25.49 | 22.38 | 21.02 |
| *Opti.* in $\mathcal{P}$+ | 36.40 | 21.63 | 120.92 | 102.61 | 78.12 | 25.57 | 22.33 | 21.02 |
| *T=50, K=25* | 8.32 | 28.36 | 40.04 | 25.68 | 85.92 | 25.47 | 21.89 | 29.04 |
| *T=50, K=50* | 8.29 | 28.37 | 39.93 | **25.66** | 85.93 | 25.45 | 21.89 | 45.04 |
| *T=10, K=10* | 25.01 | 23.89 | 85.51 | 65.81 | 81.28 | **25.64** | 22.02 | **6.79** |
| *T=100, K=10* | 35.10 | 21.70 | 119.20 | 102.36 | 78.29 | 25.61 | **22.26** | 41.23 |
| *w/o* TOPO | 8.55 | 28.18 | 41.24 | 26.48 | 85.80 | 24.46 | 20.14 | 21.02 |
| TODInv | **8.37** | **28.39** | **39.86** | 25.71 | **86.04** | 25.47 | 21.91 | 21.02 |

embedding space $\mathcal{P}$, in which all timesteps and layers of U-Net share the same optimized prompt embedding. 2) *Opti.* in $\mathcal{P}_t$, we optimize the prompt only in different timestep, in which all layers of U-Net share the same optimized prompt embedding. 3) *Opti.* in $\mathcal{P}$+, we optimize the prompt only in different layers of U-Net, and all timesteps share the same optimized embeddings. We also conduct an ablation study to investigate the effect of different sampling steps $T$ and optimization steps $K$. We develop two variants with different sampling steps $T$, $T=10$, and $T=100$, with the default optimization steps $K=10$, and develop another two variants with different optimization steps $K=25$, $K=50$ with $T=50$. Additionally, for evaluating the effectiveness of our Task-Oriented Optimization by proposing variant *w/o* Task-Oriented Prompt Optimization (TOPO) that optimizes all layers of U-Net regardless of the editing types. We conduct above ablation experiment using P2P editing on the PIE-Bench dataset.

The quantitative comparison of various variants is presented in Tab. 2. The variants *Opti.* in $\mathcal{P}$, *Opti.* in $\mathcal{P}t$, and *Opti.* in $\mathcal{P}$+ demonstrate worse performance in both structure distance and reconstruction. This suggests that optimizing prompt embeddings in these three spaces does not guarantee faithful reconstruction. Additionally, these variants show higher editability (CLIP Similarity) compared to our TODInv, as the edited images, without the constraint of source images, have more freedom to generate content according to the target prompt. In comparison, our final model, TODInv, outperforms variants *T=50, K=25* and *T=50, K=50* across all metrics, although the latter variants require more processing time. The expressiveness of the $\mathcal{P}^*$ space facilitates more effective minimization of approximation error, and 10 steps are sufficient for this process. Furthermore, both variants *T=10, K=10* and *T=100, K=10* exhibit poorer reconstruction performance. Consequently, we adhere to existing work by setting *T=50*. Compared with variant *w/o* TOPO, our final method gains the improvement in editability and reconstruction. Our task-oriented prompt optimization reduces the approximation error by optimizing prompt embeddings that are irrelevant to current editing, and achieves

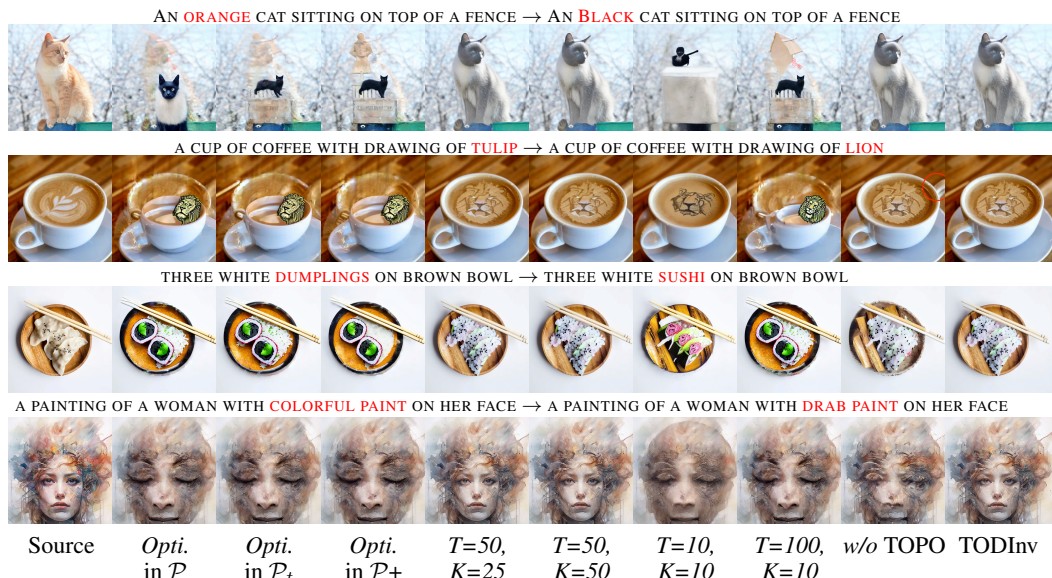

AN ORANGE CAT SITTING ON TOP OF A FENCE → AN BLACK CAT SITTING ON TOP OF A FENCE

A CUP OF COFFEE WITH DRAWING OF TULIP → A CUP OF COFFEE WITH DRAWING OF LION

THREE WHITE DUMPLINGS ON BROWN BOWL → THREE WHITE SUSHI ON BROWN BOWL

A PAINTING OF A WOMAN WITH COLORFUL PAINT ON HER FACE → A PAINTING OF A WOMAN WITH DRAB PAINT ON HER FACE

| Source | *Opti.* in $\mathcal{P}$ | *Opti.* in $\mathcal{P}_t$ | *Opti.* in $\mathcal{P}+$ | *T=50, K=25* | *T=50, K=50* | *T=10, K=10* | *T=100, K=10* | *w/o* TOPO | TODInv |

Figure 6: Qualitative comparison with various variants using P2P editing method.

better editability without influencing the reconstruction, which evidences the effectiveness of our task-oriented strategy.

We present a qualitative comparison of different variants in Fig. 6. The images edited by the variants *Opti.* in $\mathcal{P}$, *Opti.* in $\mathcal{P}_t$, and *Opti.* in $\mathcal{P}+$ show inferior results. These variants fail to preserve necessary information from the source images. In contrast, TODInv not only edits the images according to the target prompt but also maintains the unchanged parts of the image. This demonstrates the effectiveness of optimization in the $\mathcal{P}^*$ space, which preserves source information and allows for effective editing. Variants *T=50, K=25* and *T=50, K=50* yield results similar to TODInv, indicating that additional optimization steps are unnecessary for TODInv. Variant *w/o* TOPO shows structural deformation in the last sample of Fig. 6 and background perturbation in the $2_{nd}$ and $3_{rd}$ samples. With our task-oriented prompt optimization strategy, we only optimize prompt embeddings relevant to the current editing type. This approach not only reconstructs the unedited regions but also preserves editability.

### 4.6 EXTENSION ON FEW-STEP DIFFUSION MODEL

Besides the Stable Diffusion, We also extend our method on a few-step diffusion model, SDXL-Turbo (Sauer et al., 2023). We set 4 inference steps for this model, and the optimization steps $K$ is set to be 10. We compare our method with DDIM inversion, and ReNoise (Garibi et al., 2024) in the bottom rows of Tab. 1. Here we set ReNoise with the DDIM sampler for the fair comparison. We can see that our method outperforms DDIM and ReNoise both on the background preservation and CLIP similarity, with the similar inference time cost with ReNoise, which demonstrates our generalization ability on few-step diffusion model. For the qualitative comparison, please see in Appendix.

### 5 CONCLUSION AND LIMITATION

In this paper, we present TODInv, a framework that inverts and edits a real image using diffusion models tailored to specific editing tasks. We categorize various editing tasks into three types, for each kind of editing, we minimize the approximation error by optimizing specific prompt embeddings that are irrelevant to the current editing, achieving both faithful reconstruction and high editability. We conducted experiments on Stable Diffusion and SDXL-Turbo models, demonstrating the effectiveness of our TODInv over state-of-the-art methods. The primary limitation of TODInv is that it requires determining the editing types prior to inversion. However, this can be addressed by using a large language model to easily determine the types. Please refer to the Appendix for detailed instructions.

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

# A    APPENDIX

## A.1    ANALYSIS ON TASK-ORIENTED PROMPT OPTIMIZATION STRATEGY

To demonstrate the effectiveness of our task-oriented prompt optimization strategy, we present a quantitative comparison across different editing types. We evaluate variants *w/o* TOPO for appearance editing and *w/o* TOPO for structure editing. Additionally, we present the results of reversing the editing type (TODInv-Reverse), wherein appearance editing is applied to samples originally intended for structure editing and vice versa. As discussed in Sec. 4.1, the **Structure Distance** metric is not suitable for evaluating whether the images are correctly edited; therefore, we exclude this metric from the evaluation of structure editing.

The quantitative comparison is shown in Tab. 3. All variants achieve similar performance in background preservation metrics for both appearance and structure editing, as they are all optimized in the expressive $\mathcal{P}^*$ space. Our strategy optimizes prompt embeddings that are independent of the editing type, which enhances editability. Consequently, TODInv-Reverse exhibits poorer performance in CLIP similarity metrics for both appearance and structure editing compared to TODInv, which achieves the best CLIP similarity performance.

Table 3: Qualitative comparisons with various variants on different editing types.

| Variant | Editing Type | Structure | Background Preservation | | | | CLIP Similarity | |
| --- | --- | --- | --- | --- | --- | --- | --- | --- |
| | | Distance$_{\times 10^3}$ ↓ | PSNR ↑ | LPIPS$_{\times 10^3}$ ↓ | MSE$_{\times 10^4}$ ↓ | SSIM$_{\times 10^2}$ ↑ | Whole ↑ | Edited ↑ |
| *w/o* TOPO | Appearance | **8.87** | **29.16** | **38.71** | **26.75** | 86.85 | 25.44 | 23.41 |
| TODInv-Reverse | Appearance | 9.18 | 29.00 | 38.95 | 27.79 | 86.97 | 25.29 | 23.09 |
| TODInv | Appearance | 9.17 | 29.07 | 38.83 | 27.65 | **86.94** | **26.23** | **24.04** |
| *w/o* TOPO | Structure | - | 28.31 | 44.22 | 25.84 | 84.70 | 24.23 | 19.62 |
| TODInv-Reverse | Structure | - | **27.66** | 44.85 | 25.62 | 84.74 | 24.18 | 19.53 |
| TODInv | Structure | - | 28.01 | **42.49** | **24.39** | **85.07** | **25.24** | **20.63** |

We also present the qualitative comparison in Fig. 7, showing that variant *w/o* TOPO and TODInv-Reverse easily present the structure deformation. As shown in the red circle in $1_{st}$ sample of Fig. 7, variant *w/o* TOPO and TODInv-Reverse present the undesired arms in the edited images and modify the view of lions in $2_{nd}$ sample. In $3_{rd}$ sample, both variant *w/o* TOPO and TODInv-Reverse fail to preserve the facial features of source faces, and variant TODInv-Reverse also modifies the "LEGS" of children. In $4_{th}$ sample, neither variant *w/o* TOPO and TODInv-Reverse failed to remove the "FLOWER", which further demonstrates the effectiveness of our task-oriented prompt optimization strategy.

## A.2    QUANTITATIVE COMPARISON ON DIFFERENT EDITING CATEGORIES

We present the quantitative comparison on different editing categories date in Tab. 4, Tab. 5, and Tab. 6. Here we use the edited results of other methods provided by PNP's re-implementation (Ju, 2023). From Tab. 4 we can see that our TODInv outperforms other methods with P2P, MasaCtrl, and PNP editing methods on appearance editing categories on all metrics, especially on the structure preservation, our method outperforms other methods with a large step, that demonstrates the effectiveness of our TOPO strategy, by only optimizing the irrelevant layers with appearance editing, our TODInv preserves the structures information of original images effectively.

The quantitative comparison of the images with structure editing category can be seen in Tab. 5. Our TODInv outperforms other methods on all metrics with most editing methods, except with the P2P-Zero editing on background preservation, that is because P2P-Zero is proposed for image translation but not prompt-driven image editing. Compared with P2P, PNP, and MasaCtrl, DDIM and PNPInv inversion methods also receive worse performance on background preservation.

At last, the quantitative comparison of the images with global editing category can be seen in Tab. 6. Our TODInv also goes beyond other methods on most metrics.

Table 4: Qualitative comparisons on **appearance editing category** with related works using various text-guided editing methods.

| Inverse | Editing | Editing Type | Distance$_{\times 10^3}$ ↓ | PSNR ↑ | LPIPS$_{\times 10^3}$ ↓ | MSE$_{\times 10^4}$ ↓ | SSIM$_{\times 10^2}$ ↑ | Whole ↑ | Edited ↑ |
|---|---|---|---|---|---|---|---|---|---|
| DDIM | P2P | Appearance | 67.93 | 17.97 | 203.70 | 214.33 | 72.71 | 25.21 | 23.75 |
| NTI | P2P | Appearance | 14.45 | 28.10 | 55.73 | 32.46 | 85.64 | 25.77 | 24.06 |
| NPI | P2P | Appearance | 18.63 | 26.78 | 66.08 | 39.24 | 84.70 | 25.43 | 23.80 |
| StyleD | P2P | Appearance | 12.11 | 26.76 | 63.86 | 36.88 | 84.81 | 25.27 | 23.40 |
| PNPInv | P2P | Appearance | 12.39 | 28.53 | 48.22 | **27.65** | 86.39 | 25.69 | 23.93 |
| TODInv | P2P | Appearance | **9.17** | **29.07** | **38.83** | **27.65** | **86.94** | **26.23** | **24.04** |
| DDIM | MasaCtrl | Appearance | 29.09 | 22.38 | 101.20 | 84.88 | 81.03 | 24.00 | 22.20 |
| PNPInv | MasaCtrl | Appearance | 24.49 | 22.95 | 84.23 | 79.83 | 82.50 | 24.37 | **22.55** |
| TODInv | MasaCtrl | Appearance | **18.66** | **24.66** | **66.94** | **60.81** | **84.30** | **24.66** | **22.55** |
| DDIM | PNP | Appearance | 30.91 | 22.61 | 110.11 | 76.64 | 80.18 | 26.2 | 24.49 |
| PNPInv | PNP | Appearance | 26.40 | 22.89 | 104.77 | 73.82 | 81.02 | 26.21 | 24.62 |
| TODInv | PNP | Appearance | **24.22** | **25.31** | **77.87** | **54.32** | **84.13** | **27.50** | **25.43** |
| DDIM | P2P-Zero | Appearance | 74.20 | 20.21 | 169.57 | 147.82 | 76.12 | 22.95 | 21.76 |
| PNPInv | P2P-Zero | Appearance | 65.51 | **21.30** | **137.77** | **134.84** | 78.45 | 23.53 | 22.18 |
| TODInv | P2P-Zero | Appearance | **62.70** | 21.05 | 138.70 | 137.10 | **78.60** | **24.39** | **22.62** |

Table 5: Qualitative comparisons on **structure editing category** with related works using various text-guided editing methods.

| Inverse | Editing | Editing Type | PSNR ↑ | LPIPS$_{\times 10^3}$ ↓ | MSE$_{\times 10^4}$ ↓ | SSIM$_{\times 10^2}$ ↑ | Whole ↑ | Edited ↑ |
|---|---|---|---|---|---|---|---|---|
| DDIM | P2P | Structure | 17.27 | 230.19 | 237.38 | 68.45 | 24.98 | **21.33** |
| NTI | P2P | Structure | 26.30 | 69.64 | 38.70 | 82.43 | 24.23 | 20.44 |
| NPI | P2P | Structure | 25.66 | 76.98 | 41.20 | 81.82 | 24.15 | 20.54 |
| StyleD | P2P | Structure | 25.53 | 72.62 | 39.44 | 81.99 | 24.57 | 20.65 |
| PNPInv | P2P | Structure | 26.41 | 61.44 | 35.57 | 83.24 | 24.72 | 20.94 |
| TODInv | P2P | Structure | **28.01** | **42.49** | **24.39** | **85.07** | **25.24** | 20.63 |
| DDIM | MasaCtrl | Structure | 21.51 | 118.38 | 95.02 | 77.73 | 24.29 | 20.49 |
| PNPInv | MasaCtrl | Structure | 21.99 | 97.51 | 88.16 | 79.62 | 24.76 | **20.65** |
| TODInv | MasaCtrl | Structure | **23.82** | **77.82** | 66.50 | **81.36** | **25.15** | 22.49 |
| DDIM | PNP | Structure | 21.73 | 125.06 | 90.09 | 77.12 | 25.12 | 21.25 |
| PNPInv | PNP | Structure | 21.86 | 116.15 | 86.30 | 77.83 | 25.15 | 21.35 |
| TODInv | PNP | Structure | **25.04** | **82.28** | **47.75** | **81.55** | **25.48** | 20.76 |
| DDIM | P2P-Zero | Structure | 19.88 | 193.89 | 156.54 | 71.94 | 22.54 | 19.49 |
| PNPInv | P2P-Zero | Structure | **21.00** | **156.94** | **136.81** | **74.53** | 22.95 | 19.98 |
| TODInv | P2P-Zero | Structure | 20.80 | 158.12 | 150.68 | 74.27 | **23.90** | **20.03** |

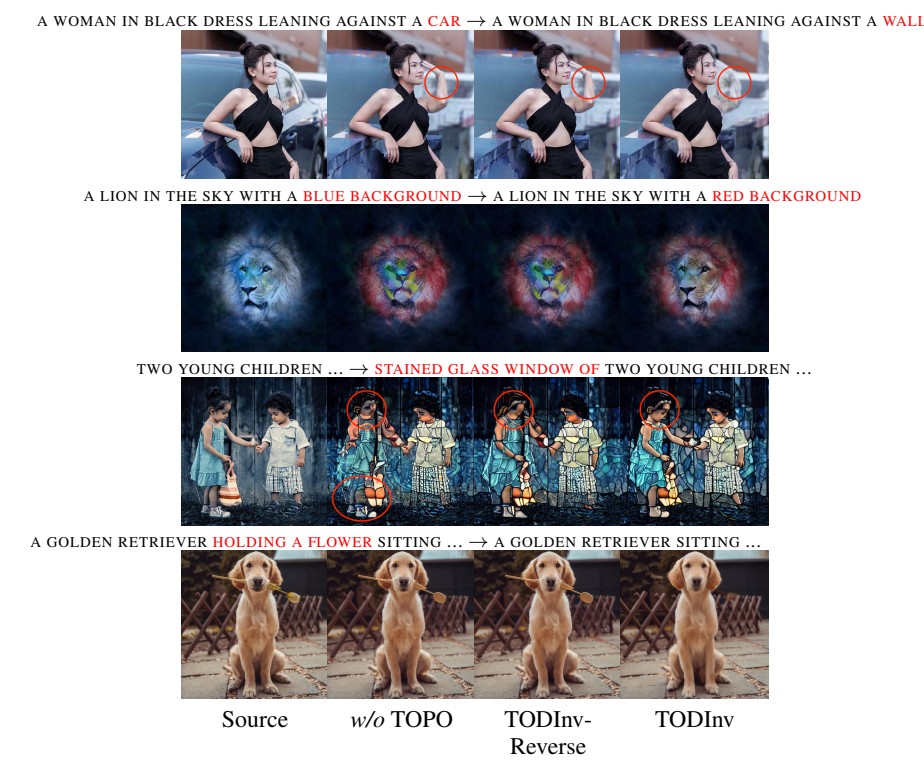

A WOMAN IN BLACK DRESS LEANING AGAINST A CAR → A WOMAN IN BLACK DRESS LEANING AGAINST A WALL

A LION IN THE SKY WITH A BLUE BACKGROUND → A LION IN THE SKY WITH A RED BACKGROUND

TWO YOUNG CHILDREN … → STAINED GLASS WINDOW OF TWO YOUNG CHILDREN …

A GOLDEN RETRIEVER HOLDING A FLOWER SITTING … → A GOLDEN RETRIEVER SITTING …

| Source | *w/o* TOPO | TODInv-Reverse | TODInv |

Figure 7: Qualitative comparison with *w/o* TOPO and TODInv-Reverse variants using P2P editing method.

Table 6: Qualitative comparisons on **global editing category** with related works using various text-guided editing methods.

| Method | | | Structure | Background Preservation | | | | CLIP Similarity | |
|---|---|---|---|---|---|---|---|---|---|
| Inverse | Editing | Editing Type | Distance$_{\times 10^3}$ ↓ | PSNR ↑ | LPIPS$_{\times 10^3}$ ↓ | MSE$_{\times 10^4}$ ↓ | SSIM$_{\times 10^2}$ ↑ | Whole ↑ | Edited ↑ |
| DDIM | P2P | Global | 66.97 | 19.12 | 165.37 | 185.68 | 75.70 | 24.78 | **23.02** |
| NTI | P2P | Global | 16.56 | 27.50 | 48.43 | 34.37 | 86.10 | 24.40 | 21.69 |
| NPI | P2P | Global | 17.80 | 26.93 | 53.82 | 36.87 | 85.73 | 24.42 | 21.98 |
| StyleD | P2P | Global | 14.44 | 26.54 | 53.52 | 38.47 | 85.33 | 24.49 | 21.64 |
| PNPInv | P2P | Global | 12.58 | 27.80 | 45.03 | 31.73 | 86.62 | 24.68 | 22.00 |
| TODInv | P2P | Global | **9.48** | **28.59** | **34.90** | 26.83 | **87.40** | **25.89** | 21.62 |
| DDIM | MasaCtrl | Global | 25.61 | 23.45 | 85.26 | 70.79 | 82.75 | 23.15 | 21.10 |
| PNPInv | MasaCtrl | Global | 22.52 | 23.79 | 69.85 | 66.34 | 84.07 | 23.54 | 21.12 |
| TODInv | MasaCtrl | Global | 19.39 | 25.29 | **55.96** | 54.13 | **85.23** | 23.90 | 22.86 |
| DDIM | PNP | Global | 29.69 | 23.20 | 90.48 | 75.78 | 82.32 | 24.90 | **22.57** |
| PNPInv | PNP | Global | 27.09 | 23.38 | 84.53 | 73.56 | 82.56 | 24.81 | 22.51 |
| TODInv | PNP | Global | **26.74** | **25.17** | **70.53** | 51.64 | **84.47** | **25.45** | 22.06 |
| DDIM | P2P-Zero | Global | 57.89 | 21.92 | 125.83 | 112.53 | 79.43 | 23.16 | 21.10 |
| PNPInv | P2P-Zero | Global | 42.69 | **22.93** | 99.60 | 98.70 | **81.40** | 23.80 | **21.81** |
| TODInv | P2P-Zero | Global | **43.25** | 22.84 | **98.12** | **96.21** | 81.27 | **24.54** | 21.50 |

## A.3 QUANTITATIVE COMPARISON ON SDXL-TURBO

We present the qualitative comparison in Fig. 8, we can see that ReNoise introduces the unnecessary structure deformation, and our method captures the source structure effectively.

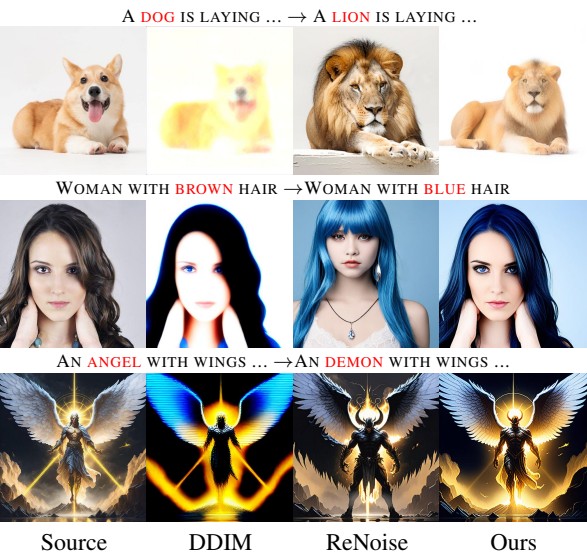

A DOG IS LAYING ... → A LION IS LAYING ...

WOMAN WITH BROWN HAIR →WOMAN WITH BLUE HAIR

AN ANGEL WITH WINGS ... →AN DEMON WITH WINGS ...

Source          DDIM          ReNoise          Ours

Figure 8: Qualitative comparison on SDXL-Turbo.

## A.4 MORE QUALITATIVE COMPARISON WITH MASACTRL, PNP, AND P2P-ZERO EDITING METHODS

The qualitative comparison based on MasaCtrl, PNP, and P2P-Zero editing methods are shown in Fig. 9, Fig. 11, and Fig. 10 respectively.

As shown in the $2_{nd}$ sample of Fig. 11, all competitors fail on local appearance editing. In the $5_{th}$ sample, none of the competitors capture the editing instruction of "A BLACK AND WHITE SKETCH", and pay more attention to "PINK" incorrectly. The same problem also emerged on modifying the "RED DRINK" to "RED WINE". That evidences the effectiveness of our method in capturing semantic instruction.

As shown in the red cycles in $1_{st}$ sample of Fig. 9, most of competitors can not preserve the chains in the original image. Our TODInv is also skilled at object removal rainbow.

In Fig. 10, DDIM and PNPInv fail to preserve face details when editing the "SHIRT" to "SWEATER", and they also failed to preserve the color of the bear. Our TODInv preserves more source details during editing. That should contribute to our task-oriented strategy, as we optimize the prompt embeddings that are irrelevant to the current editing, which preserves the source details effectively.

## A.5 THE ALGORITHM OF TODINV

The algorithm of our TODInv inversion and editing can be seen in Alg. 1.

## A.6 EDITING TYPE DETERMINATION

As discussed in Sec. 5, the main limitation of ToDInv is we need to determine the editing type before inversion. It may be not easy for unprofessional users. However, it is easy to determine the edit type based on the source and target prompts using ChatGPT. We present the illustration of determining editing types with ChatGPT in Fig. 12, we can see that it is easy to determine the editing type with our given prompts.

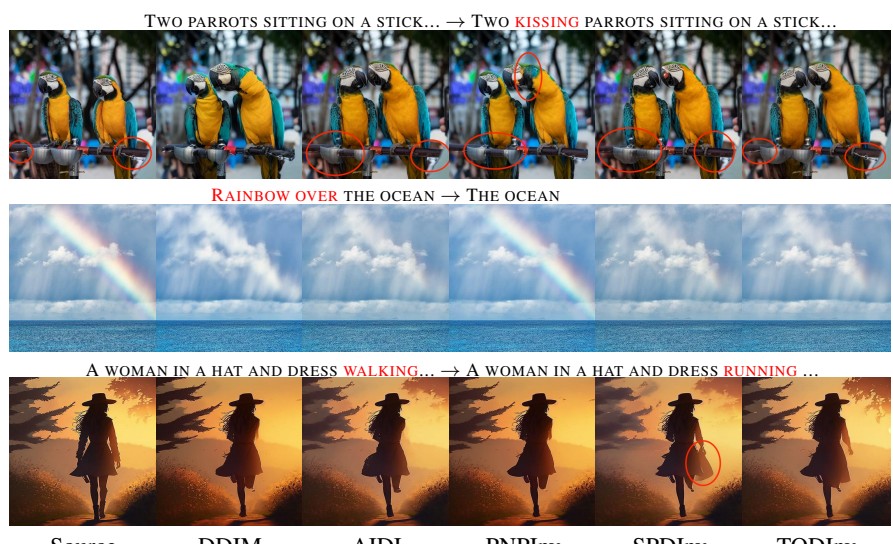

Figure 9: Qualitative comparison with various inversion methods using MasaCtrl editing method.

---

**Algorithm 1:** Algorithm of TODInv.

---

**Part I : Inversion Pipeline**

**Input:** Source image latent $z_0$, DDIM steps $T$, source prompt embedding $P$, maximal
    optimization step $K$, threshold $\delta$, Editing type `Type`.

**Output:** Latent noise $z_T$, Optimized prompt embedding in each timestep $P_t^*$.

1: **for** $t \leftarrow 1$ to **T do**
2:     Get $z_t$ from $z_{t-1}$ using DDIM inversion (Eq. 5);
3:     **for** $i \leftarrow 0$ to **K do**
4:         Initialize the current prompt embedding $P_t$ as $P$;
5:         Update $z_t'$ using $z_t$ and $P_t$ (Eq. 8);
6:         Optimize specific layers of $P_t^*$ (determined by `Type`) by minimizing $\|z_t - z_t'\|_2^2$ (Eq. 9);
7:         **if** $\|z_t - z_t'\|_2^2 < \delta$ **then** *Break* **end if**
8:     **end for**
9: **end for**

---

**Part II: Reconstruction and Edit Pipeline**

**Input:** Target prompt embedding $P^{target}$, latent noise $z_T$, optimized prompt embedding in each
    timestep $P_t^*$, text-guided image editing method $E$.

**Output:** Reconstructed latent $z_0^r$, Edited latent $z_0^e$.

1: **for** $t \leftarrow T$ to $0$ **do**
2:     Update the reconstructed latent $z_t^r$ based $z_T$ and $P_t^*$ using DDIM sampler;
3:     Renew the target prompt embedding $\tilde{P}_t^{target}$ using $P_t^*$ and $P_t^{target}$ (Eq. 10);
4:     Update the edited latent $z_t^e$ based $z_T$ and $\tilde{P}_t^{target}$ using $E$,
5: **end for**

---

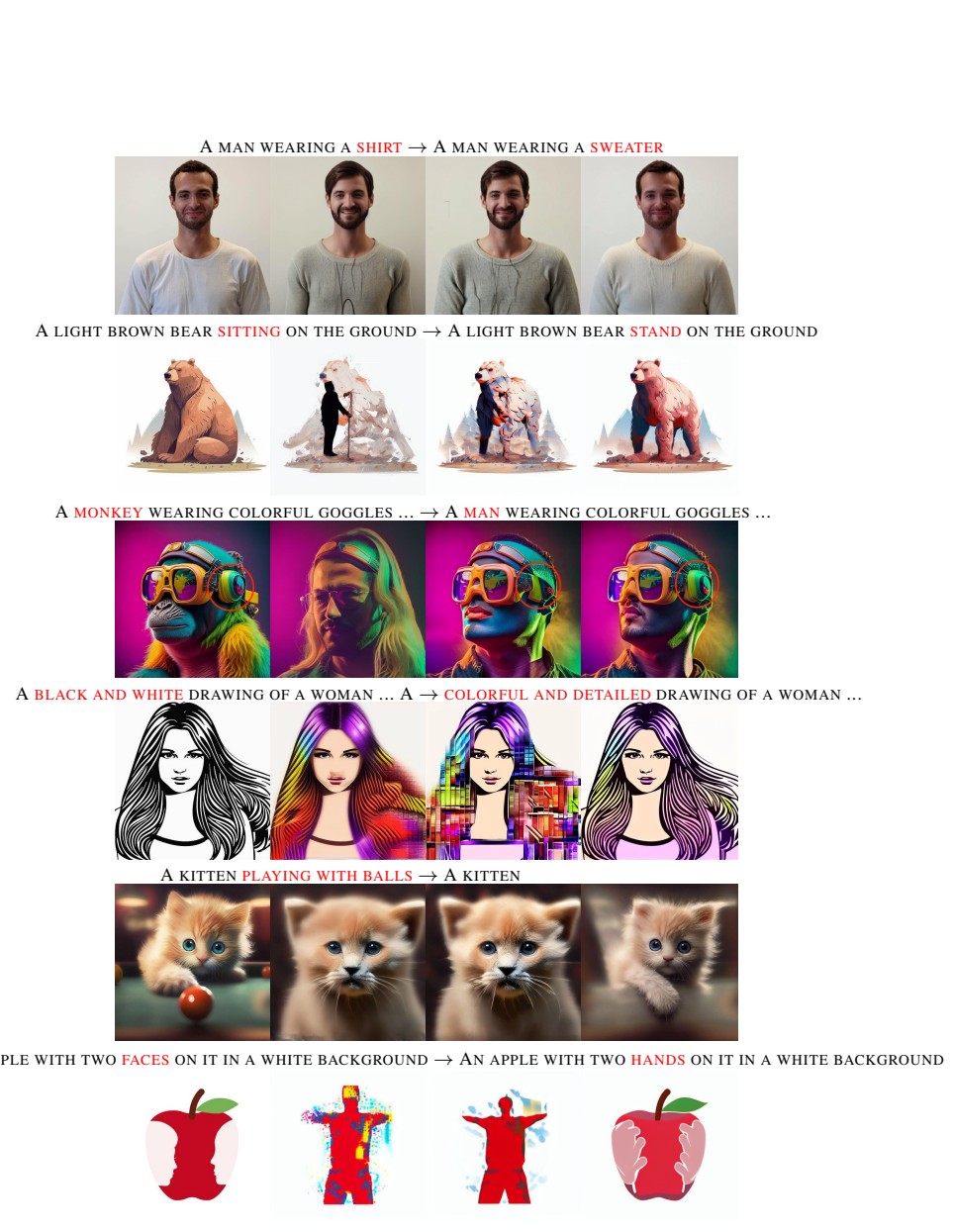

Figure 10: Qualitative comparison with various inversion methods using P2P-Zero editing method.

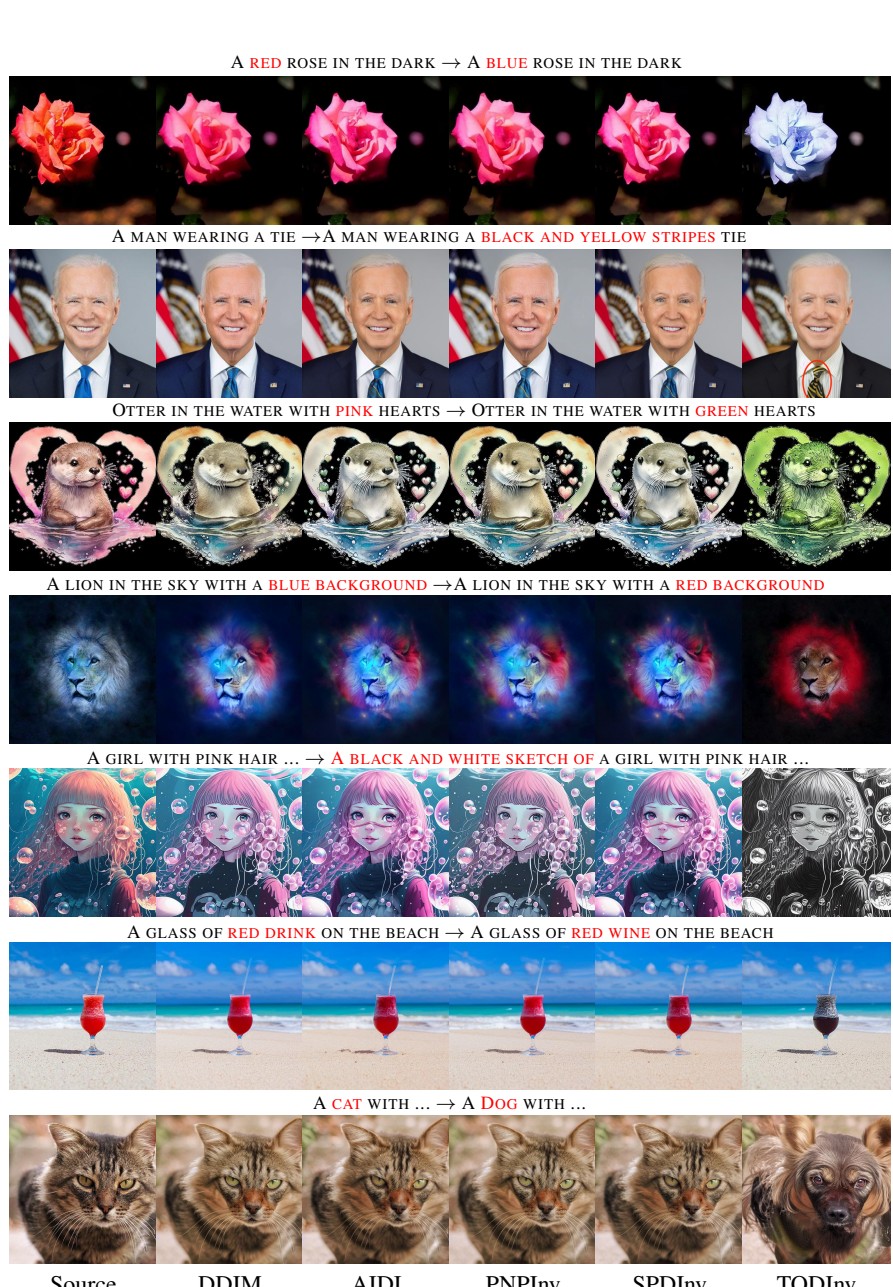

Figure 11: Qualitative comparison with various inversion methods using PNP editing method.

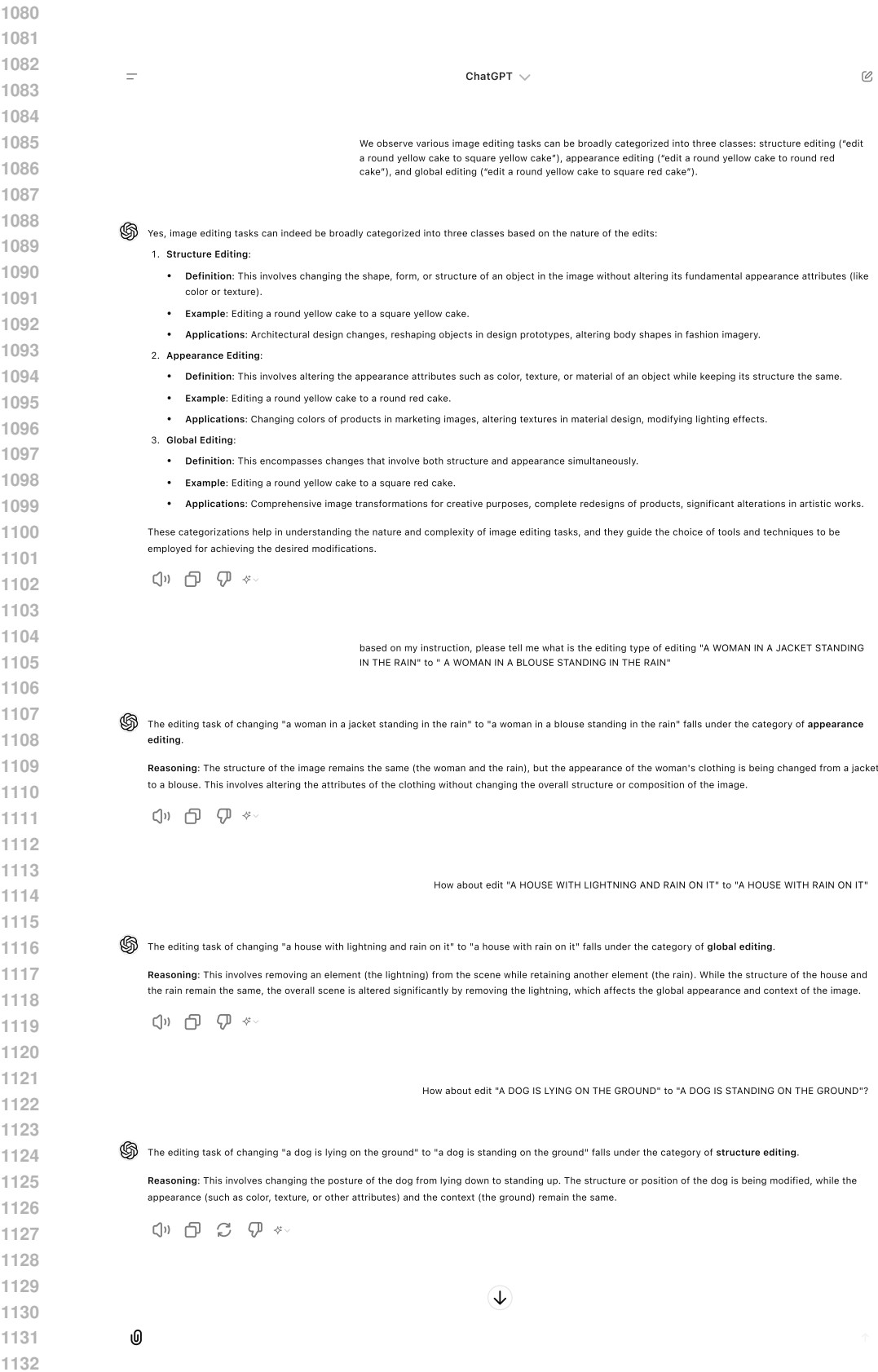

Figure 12: Illustration of determining editing types with ChatGPT.

