# OpenReview forum: "Task-Oriented Diffusion Inversion for High-Fidelity Text-based Editing"
_ICLR.cc/2025/Conference — ICLR 2025 Conference Withdrawn Submission_

### Official Review · Reviewer_cMDx · 2024-10-29

**Soundness:** 3
**Presentation:** 2
**Contribution:** 2
**Rating:** 5
**Confidence:** 4

**Summary:**

This work points out that existing diffusion inversion methods achieve good reconstruction but limited editability due to their separate treatment of inversion and editing processes. To address this limitation, the authors propose a new diffusion inversion method, TODInv, for high-fidelity text-based real image editing.

The authors first categorize text-based image editing prompts into three classes: structure editing, appearance editing, and global editing. Inspired by the extended prompt embedding space P*, they observe that different layers in the U-Net architecture exhibit varying levels of importance depending on the editing class. Consequently, they perform diffusion inversion through prompt optimization in P* space only on the relevant layers for each editing class. They demonstrate the effectiveness of TODInv through extensive quantitative and qualitative experiments.

**Strengths:**

1. The paper is well-motivated: Even within text-based image editing tasks, different editing prompts require distinct editing capabilities, making it challenging to faithfully execute all editing prompts with a single unified approach. This paper effectively identifies this limitation and proposes a new method based on this observation.
2. The method is simple and the paper is easy to follow.
3. The paper provides extensive quantitative and qualitative experimental results across various editing methods and inversion methods.

**Weaknesses:**

1. Poor presentation
    * The paper contains several errors in highlighting the best values in bold within tables. Most of these errors occur when the proposed method's values are incorrectly marked in bold despite not being the best performing results. These errors could potentially mislead readers into overestimating the performance of the proposed method and significantly diminish the credibility of the reported experimental results.
    * There is a significant formatting error in [page 7, line 343] where text lines overlap. While such formatting issues might seem minor, given that the submission has reached the 10-page limit, resolving this overlap would likely cause the paper to exceed the page limit - a violation of ICLR's submission guidelines.
2. Limitations of proposed methods
    * The paper's core concept of "task-oriented" begins with classifying given editing prompts, but this classification relies on ChatGPT. However, the paper lacks quantitative analysis regarding ChatGPT's classification performance. Moreover, this dependency should be accounted for in the inference time measurements.
    * The method's requirement to pre-define class-specific editing layers within the U-Net architecture introduces architectural constraints that limit its broader generalizability.
3. The lack of implementation code limits reproducibility and credibility of the reported experimental results.

**Questions:**

1. Why does the method exhibit inferior performance at T=10 and T=100 compared to T=50? This observation appears counterintuitive, as diffusion models typically demonstrate improved performance with an increasing number of timesteps.
2. To substantiate the claim that ChatGPT can perform editing task classification, a quantitative analysis demonstrating its classification accuracy on the PIE-Bench dataset images using the ChatGPT API is necessary.
    * Given the high operational costs associated with the ChatGPT API, exploring the use of open-source MLLMs presents a viable alternative. Additionally, fine-tuning a smaller model on task-specific datasets for editing classification could potentially offer a more cost-effective solution while maintaining performance.
3. Can we use TODInv on DiT [1] style models too? Recent emergence of DiT architectures as alternatives to UNet raises questions about TODInv's potential applicability to DiT models.
4. Why do the structural similarity results exhibit notably strong performance for P* while demonstrating significantly weaker performances for P, Pt, and P+ in Table 2?

[1] Scalable Diffusion Models with Transformers

---

### Official Review · Reviewer_WnpS · 2024-11-02

**Soundness:** 2
**Presentation:** 3
**Contribution:** 2
**Rating:** 5
**Confidence:** 4

**Summary:**

The paper introduces Task-Oriented Diffusion Inversion (TODInv), a framework designed to enhance text-based editing capabilities for real images. TODInv operates by optimizing prompt embeddings within the extended P* space, utilizing distinct embeddings across various U-Net layers and time steps. The framework categorizes editing tasks into structure, appearance, and global edits, optimizing embeddings unaffected by the current editing task.

**Strengths:**

- The idea is straightforward.
- The proposed TODInv performs well on certain single object editing examples, editing flexibility and background preservation are balanced.
- By further categorizing the editing types into structure/appearance/structure & appearance, more fine grained qualitative and quantitative analysis are carried out on PIE-bench.

**Weaknesses:**

- The idea of extended textual inversion is not new (e.g. neti), the extension proposed in this paper seems less flexible and heuristic.
- The inversion and editing paradigm in Fig. 3 is not clearly explained by the caption. A dedicated paragraph to explain the process as well as its relation to previous editing pipeline should be added.
- The lacks evaluations in more complex scenarios such as multiple objects and larger area editing.
- Requires more manual setting which could lead to less stable performance in different scenarios.
- To demonstrate the robustness of the proposed method, as well as the soundness of the claims, additional ablation experiments could be helpful, such as:
    * changing the ranges/numbers of appearance and structure layers (to demonstrate the robustness)
    * changing the editing area scales for structure editing, (if the editing area is small, even though there is structure change, should it be appearance editing? If so the editing type should be categorized by editing area.)
    * using miss matched editing types, (to show the soundness of the proposed method)
- Minor formatting issues, e.g.,
    * text overlapping in line 342.
    * Time (s) not Times (s) in Table 1

**Questions:**

- How are the editing types chosen on the qualitative experiments on PIE-Bench? Are they set manually or automatically labelled as described in A.6?
- What are the connections between the editing methods and the inversion methods according to the experiments? Should the inversion implementation (the number and positions of structure and appearance layers) change for different editing methods?

**Details Of Ethics Concerns:**

NA.

---

### Official Review · Reviewer_Lbsd · 2024-11-02

**Soundness:** 2
**Presentation:** 3
**Contribution:** 1
**Rating:** 3
**Confidence:** 4

**Summary:**

TODInv is a finetuning required  text-guided image editing method, which categorized the image editing task into three subtasks, structure editing, appearance editing and global editing. For each class of editing, TODInv minimizes the approximation error by optimizing specific
prompt embeddings that are irrelevant to the current editing. TODInv  optimizes the prompt embedding in the previously well-studied extended p-star space for this three different tasks. Extensive quantitative experiments are conducted on PIE-bench, which demonstrated its effectiveness.

**Strengths:**

1. The quantitative experiments are dense, which is good.

2. The network structure figures are drawn in a professional way.

3. The method is explained clearly and the paper is easy to follow.

**Weaknesses:**

1. The quality of the edited images are very poor compared with SoTA methods nowadays, considering that it is the end of 2024 now.  The techniques in this paper are still from almost two  years ago, which have been improved a lot in recent years.

2. The method requires optimization. Yet its editing results are far worse than Imagic from CVPR 2022 and its later improved versions. The authors did not compare with Imagic and the later methods improving Imagic. Especially, Imagic and its improved versions are capable of conducting various non-rigid editing well, which TODInv should compare with in the TEdBench benchmark.

3. The separated prompt embedding optimization according to different editing task is a waste of time considering the fact that there are many unified text-guided image editing methods already capable of conducting all image editing tasks well, including structure editing, appearance editing and global editing.

4. This paper severely lacks novelty. The extended prompt embedding space has been well studied in 2023. The various disentangled properties of UNets have been thoroughly explored by many works in recent two years. The optimization techniques in this paper are disused nowadays.


Overall, unfortunately, the quality and novelty of this paper is too far from ICLR standard.

**Questions:**

please refer to the weaknesses.

---

### Official Review · Reviewer_Audw · 2024-11-03

**Soundness:** 2
**Presentation:** 3
**Contribution:** 2
**Rating:** 3
**Confidence:** 3

**Summary:**

The authors proposed a framework that jointly optimizes inversion and editing processes within the extended $P^∗$ space, which is using distinct embeddings across different U-Net layers and time steps.

**Strengths:**

-	The inversion method seems to perform well on multiple editing models.
-	The method is clearly descripted.

**Weaknesses:**

-	I have some concerns about the technical contribution of the paper. The main contribution claimed is the joint optimization process of inversion and editing framework. However, my understanding is that it only contains the inversion process that tries to optimize $P^*$. The second claimed contribution is the idea of classing different layer prompt embeddings into multiple groups according to the resolution. And this is adopted from a personalized editing work NeTI [1]. In general, I don’t think technical contributions of the paper meet the bar of ICLR.
-	The paper does not appear to achieve superior results compared to existing methods. The **qualitative** results (i.e., fig.1& 5) are similar to those from PNPInv, and SPDInv results are not so convincing to me. The results of SPDInv are largely unchanged in many cases (see fig.1& 5), which needs a reasonable explanation. Additionally, the paper lacks separate comparative evaluations for appearance, structure, and combined editing, as it claims task-oriented editing.

[1] Alaluf, Yuval, et al. "A neural space-time representation for text-to-image personalization." ACM Transactions on Graphics (TOG) 42.6 (2023): 1-10.

**Questions:**

-	Is there any evidence supporting the decision to class layer prompt embeddings into two groups based on resolution: the structural prompts in low-resolution layers and appearance prompts in high-resolution layers? It would be helpful to adding experiments to illustrate this.
-	How to define or optimize the fixed $p$ (in fig. 4) that not belong to structure and appearance prompt set?
-	Minor issues: typesetting problem in line 343; overlapping elements in fig 3.

---

### Official Review · Reviewer_VHwx · 2024-11-04

**Soundness:** 2
**Presentation:** 2
**Contribution:** 2
**Rating:** 5
**Confidence:** 5

**Summary:**

The paper proposes task-oriented prompt optimization for editing.

**Strengths:**

The authors proposes layer-wise prompt optimization for inversion

**Weaknesses:**

1. The paper lacks novelty. The idea of layer-wise optimization comes from P+:Extented prompt optimization paper. Although P+ is extension from textual inversion, i think the proposed inversion is just simple modification of Null-text inversion with P+.

2. The method lacks efficiency. It requires much time for optimization, which takes over 20 seconds. Since there are so many editing methods which enables editing within 5 seconds or even using single step, this proposed method has no clear advantage compared with those recent methods.

3. Although the method shows comparison between different inversion methods, the primary point for editing quality evaluation comes from subjective metrics. Please show some result on user study.

**Questions:**

See weakness

---

### Note · Authors · 2024-11-12

I have read and agree with the venue's withdrawal policy on behalf of myself and my co-authors.